# Heterogeneous Condition of Asthmatic Children Patients: A Narrative Review

**DOI:** 10.3390/children9030332

**Published:** 2022-03-01

**Authors:** Cristiano Caruso, Stefania Colantuono, Stefania Arasi, Alberto Nicoletti, Antonio Gasbarrini, Angelo Coppola, Loreta Di Michele

**Affiliations:** 1Fondazione Policlinico Universitario “A. Gemelli” IRCCS, Dipartimento di Scienze Mediche e Chirurgiche, Università Cattolica del Sacro Cuore, 00168 Rome, Italy; stefania_colantuono@hotmail.it; 2Translational Research in Pediatric Specialities Area, Division of Allergy, Bambino Gesù Children’s Hospital, IRCCS, Piazza Sant’Onofrio, 00165 Rome, Italy; stefania.arasi@yahoo.it; 3CEMAD Digestive Disease Center, Fondazione Policlinico Universitario “A. Gemelli” IRCCS, Catholic University of Rome (Italy), 00168 Rome, Italy; alberto.nicoletti@unicatt.it (A.N.); antonio.gasbarrini@unicatt.it (A.G.); 4Division of Respiratory Medicine, Ospedale San Filippo Neri-ASL Roma 1, UniCamillus, Saint Camillus International University of Health Sciences, 00131 Rome, Italy; coppolangelo@gmail.com; 5Pulmonary Interstitial Diseases Unit, UOSD Interstiziopatie Polmonari Az Osp. S. Camillo-Forlanini, 00100 Rome, Italy; dottoressadimichele@gmail.com

**Keywords:** sleep disorders, childhood asthma, adolescent asthma, severe asthma, microbiome, biomarker

## Abstract

Currently, asthma represents the most common chronic disorder in children, showing an increasingly consistent burden worldwide. Childhood asthma, similar to what happens in adults, is a diversified disease with a great variability of phenotypes, according to genetic predisposition of patients, age, severity of symptoms, grading of risk, and comorbidities, and cannot be considered a singular well-defined disorder, but rather a uniquely assorted disorder with variable presentations throughout childhood. Despite several developments occurring in recent years in pediatric asthma, above all, in the management of the disease, some essential areas, such as the improvement of pediatric asthma outcomes, remain a hot topic. Most treatments of the type 2 (T2) target phenotype of asthma, in which IL-4, IL-5, and IL-13 modulate the central signals of inflammatory reactions. Although, there may be an unresolved need to identify new biomarkers used as predictors to improve patient stratification using disease systems and to aid in the selection of treatments. Moreover, we are globally facing many dramatic challenges, including climate change and the SARS-CoV2 pandemic, which have a considerable impact on children and adolescent asthma. Preventive strategies, including allergen immunotherapy and microbiome evaluation, and targeted therapeutic strategies are strongly needed in this population. Finally, the impact of asthma on sleep disorders has been reviewed.

## 1. Introduction

### Overview on Pediatric Asthma

Asthma represents a very common condition in childhood and adolescence. A goal of precision medicine is to define strategies of treatment in this population. Shared decision making and self-management are some aspects of this process to minimize the impact of asthma in real-life [1]. The importance of biomarkers related to the inflammatory status of the airways represents a hot topic of research in both adults and children. To date, biomarkers to identify the right patient and related therapy for both screening and follow-up have not been identified in normal clinical practice, and this is the main unmet need for precision medicine in the coming years [2]. Asthma and in particular pediatric asthma is composed of various factors that influence the evolution of the disease [3]. Finally, in recent years, research has begun to focus on the pediatric population, on phenotyping, and on clustering groups of patients on the basis of different clinical characteristics and inflammatory phenotypes. Compared to onset in adulthood, childhood-onset asthma disease is usually characterized by elevated levels of total immunoglobulin E (IgE), blood eosinophil count and multiple sensitization to aeroallergens. Asthma in adolescents and children is statistically correlated with the need for specialist visits and as a cause of hospitalizations and access to the emergency room (ED). Until recently, there were no major specific correlates between disease and severity.

## 2. Diagnostic Tools

### Biomarker for Diagnosis and Follow-Up

The new drugs indicated in type 2 inflammation act on mast cells, eosinophils, and cytokines IL-4, IL-5, and IL-13 and find their target in immune-mediated reactions. Unfortunately, stratification of patient type and corresponding drug of choice is currently limited to the use of blood eosinophil counts or sputum, fractionated exhaled nitric oxide (FENO) or blood protein markers such as total IgE or periostin, which do not provide sensitive and specific data. For this reason, there remains an unsatisfied need to identify new predictive biomarkers of disease to improve the clustering of patients through physiopathological mechanisms and to help precision medicine.

In a recent article, Komert et al. [4] confirmed that the study of urinary metabolites and LTE4 PGD2 can be considered a non-invasive marker of T2 inflammation. The use of eicosanoids could be considered as a biomarker due to the fact that they have been shown to be potent and biologically active as inflammatory mediators and signals for the activation of particular inflammation cells, in particular eosinophils and mast cells. For this reason, it is legitimate to think of using urinary eicosanoids for stratification of patients for treatment with biological agents that target T2 inflammation (for example, anti-IL-5, anti-IL4Ra, and anti-alarmine). In another recent study [5] it was noted that periostin is a good marker of eosinophilic inflammation in patients with asthma. Moreover, for this reason, some studies have focused on periostin levels of exhaled breath condensate (EBC) in pediatric patients with asthma as a potential biomarker for the disease [6]. Unfortunately, the low periostin level in patients with mild asthma and the lack of differences between asthmatic subjects and controls indicate that EBC periostin may not be useful as an asthma biomarker in this group.

In a recent study [7] the correlation between obesity and asthma in patients with adiponectin levels and resistin/adiponectin and leptin/adiponectin ratios was demonstrated. It has been noted that blood levels of YKL-40 and resistin may be associated with inflammation of the airways. The study of Shah et al. evaluated the role of the eosinophilic cationic protein in the pediatric population [8]. Plasma ECP concentrations were always associated with type 2 inflammation biomarkers. At baseline, children in the higher ECP tertile had poor disease control, greater airflow limitation, and more exacerbations, but they also had poor disease control, and a more effective therapeutic response with intramuscular triamcinolone. At 12 months, associations between higher ECP tertile and exacerbations, but not lung function or asthma control, persisted after covariate adjustment. Unfortunately, the sensitivity and specificity of the ECP was low and was not significantly different from that of the blood eosinophil count.

## 3. Transitional Care

The age of adolescents and young adults (AYA) face specific needs and challenges ranging from psychological and medical fields to educational and vocational ones. To address the specific needs of patients at this particular age in a developmentally appropriate way is the aim of transitional care. This specific period of life is challenging because of the numerous life events occurring and linked with changes in terms of education, establishment of relationships, and even work and sometimes new places. The perception of the symptoms and the understanding of the consequences of diseases change in this specific life-time. AYA may fail to be responsible for self-management, with a suboptimal/worse adherence to treatment. Furthermore, many AYA move from pediatric to adult health services, and they may perceive a sense of loss and fear of the unknown that may lead to loss of confidence in health professionals, failure to follow up, and lack of compliance. Therefore, the consequences impact on the symptoms they experience, their management and even on how the symptoms and management are perceived and accepted. Specific training is often needed to make them more confident with their own disease.

The European Academy of Allergy and Clinical Immunology (EAACI) has provided evidence-based recommendations to support the transitional care of allergic/asthmatic AYA [9], including useful procedures such as the simplification of medication regimes and the use of reminders or peers in training AYA patients. The involvement of families and friends should be encouraged in assisting patients; letting their friends know about their allergies and asthma could improve disease acceptance [10].

## 4. Specific Current Challenges

In this specific era, we are globally facing many dramatic challenges including climate change and the SARS-CoV2 pandemic.

The Center for Disease Control (CDC) has defined the targets of climate as air pollution, vector-borne diseases, allergens, water quality, water and food supply, environmental deterioration, extreme heat, and severe weather. All of these changes represent a serious threat to health [11].

According to the Environmental Protection Agency (EPA), warmer temperatures may impair air quality and increase the concentrations of air and water pollutants [12] with detrimental repercussions on respiratory health by directly causing or aggravating pre-existing respiratory diseases and increasing exposure to risk factors for respiratory diseases, such as asthma [13]. Global warming leads to early flowering, which increases and prolongs the concentration of pollen in the air. Moreover, exposure to stronger amounts of pollen and mold may make people that do not currently have allergies develop allergic symptoms [14]. The ground-level ozone causes airway inflammation and damages lung tissue [13]. Children, together with the elderly, people with lung disease, and those who are actively outdoors, represent the most vulnerable subjects to these changes. Emergency room visits and hospitalizations of asthmatic people are often subsequent to increasing ground-level ozone pollution [14].

Alongside climate change, in the last couple of years we have been experiencing the COVID-19 pandemic. The spread of strict protective measures, including social distancing, mask wearing, and home quarantine, has been implemented to reduce SARS-CoV-2 transmission. As asthma exacerbations can be triggered by respiratory infections and air pollution, among others [15,16], COVID-19 and its resulting social isolation measures have effects on chronic airway disease patients in several ways. Patients with asthma have experienced improved disease control during the pandemic due to behavioral changes, including reduced exposure to asthma triggers and increased treatment adherence, considering also that social distancing and facemasks use may decrease viral circulation [17]. Furthermore, the decreased frequency of acute episodes does not support the notion that childhood asthma may be a risk factor for COVID-19 [18].

## 5. Preventive Strategies

### 5.1. Preventive Allergy Immunotherapy

Prevention is one of the major concerns, mainly in early ages. From this point of view, allergen immunotherapy (AIT) has preventive effects, representing a disease-modifying treatment for IgE-mediated allergic disease with effects beyond cessation of AIT. EAACI has developed a clinical practice guideline to provide evidence-based recommendations for AIT in the context of the prevention of allergic diseases [19]. Moderate-to-severe allergic rhinitis (AR) patients, triggered by grass/birch pollen allergy, are eligible for a 3-year course of subcutaneous or sublingual AIT, that prevents asthma for up to 2 years post-AIT in addition to the effects on allergic rhinitis symptoms and medication [20]. Although there is evidence for a longer preventive potential of AIT, suggested also by some clinical trials [21], more high-quality studies are awaited.

### 5.2. Microbiome and Airway Inflammation

A large body of evidence has highlighted the crucial role of the interaction between the immune system and the microbiota in the development of immune-mediated diseases. Both the gut, the upper-airways, and the environmental microbiota have been studied in order to identify specific imbalances that may be associated with asthma.

A higher α-diversity and a difference in β-diversity of the microbiota in the upper-airways is a predisposing factor for the development of asthma in children. In addition, a prevalence of Veillonella and Prevotella in the first month of life has been correlated with the development of asthma within age six years [22].

Children with a predominance of Staphylococcus in the nasopharyngeal mucus microbiota during the first 6 months of life have a higher risk of recurrent wheezing and asthma. The detection of rhinovirus and a Moraxella-dominated microbiota during episodes of respiratory illnesses is associated with asthma that persists during late childhood [23].

On the other hand, children with asthma whose microbiota was dominated by the presence of Corynebacterium and Dolosigranulum had significantly lower risk of early loss of asthma control, exacerbations, and rhinovirus infection. Moreover, a shift from a Corynebacterium and Dolosigranulum-dominated microbiota to a Moraxella-predominant one is correlated with an increased risk of severe asthma exacerbations [24]. As for the gut microbiota, some studies observed that a reduction in the microbiota diversity in early infancy predicted the development of asthma at school age [25]. In another study, a decrease in the abundance of specific intestinal species, such as Faecalibacterium, Lachnospira, Veillonella, and Rothia, in the first 100 days of life, has been associated with childhood asthma [26]. However, the composition of the gut microbiota at preschool age (2–4 years) is no longer associated with asthma [27]. Taken together, this evidence highlights that the influence of the gut microbiota may be higher during early infancy and decreases with time.

In addition, environmental microbiota may play a role in children’s asthma development. In particular, a higher bacterial diversity in the mattress dust of children who grew up on a farm inversely correlates to the development of asthma [28]. In conclusion, a better understanding of the interaction between the microbiota and the immune system may allow a way to prevent and treat asthma in children to be identified.

## 6. Therapeutic Approach to Asthma and Severe Asthma in Childhood and Adolescence

Nowadays, asthma represents the most common chronic disease in children, showing an increasingly consistent burden worldwide [29].

Childhood asthma, similar to what happens in the adult, is a heterogeneous disease with a large variability of phenotypes, according to the genetic background of patients, age, severity of symptoms, risk factors, and comorbidities [30], and cannot be considered a singular well-defined disease, but rather a uniquely assorted disorder with variable presentations throughout childhood [31]. We must also underline that the role of functional tests, particularly spirometry, can also be very variable in children. Although not the subject of discussion in this review, we consider very relevant the fact that respiratory functional tests often find little use in childhood, and even though the diagnosis should be based on both clinical and spirometric evaluation, in the primary care setting and in pediatric setting, the use of spirometry must be encouraged [32].

Therefore, like in the management of asthma in the adult patient, asthmatic children or adolescents need a regular monitoring to keep good asthma control, prevent asthma attacks, and manage comorbidities, considering also that in order to manage the disease during childhood, parents need to realize asthma as an illness, understand the treatment of asthma, and be able to monitor and respond to changes in the context of family life, leading their asthmatic child to the development of self-management responsibility [33].

Despite several developments occurring in recent years in pediatric asthma, above all in the management of the disease, some essential areas such as the improvement of pediatric asthma outcomes, remain a hot topic [34].

In the western society asthma remains a leading cause of chronic disease in children with a high prevalence of incidents [35]. Effective medications, particularly the inhaled corticosteroids (ICS), are currently the cornerstone in asthma treatment in children but a substantial proportion of them (40%–70%) reach only a suboptimal control of the disease [36,37].

The Global Initiative for Asthma (GINA) 2021 report has introduced landmark changes for the improvement of asthma prevention and management also in childhood and adolescence [38].

According to this document, it is underlined that the risk of exacerbation is reduced by inhaled corticosteroid-containing treatment and also by avoiding short-acting β₂-agonist-only treatment. Particularly, at any level of asthma severity, a reliever treatment that includes as-needed inhaled corticosteroid–formoterol is recommended [39].

However, it remains controversial if a one-size-fits-all approach can work better than a step-up and step-down scheme according to disease control, because the heterogeneity of asthma requires specific and more tailored therapeutic strategies and monitoring [30]. Compared to normal weight children, for example, obese children showed a worse response to ICS in terms of lung function and exacerbations [40]. Moreover, in the management of children or adolescent patients with asthma, the transitional process during the different phases of the disease must be taken into account. Transition is an ‘active and evolving process that addresses the medical, psychosocial, and educational needs of young people as they prepare to move from child- to adult-centered health care’ [41]. Transition programs have shown significant improvements in the management of patients with chronic diseases, reaching a greater level of adherence in follow up visits, and of self-efficacy outcomes [10].

According to the heterogeneity of asthma, several patient characteristics and respiratory function parameters and biomarkers can predict the response to inhaled corticosteroids (ICS), montelukast, or long-acting beta-agonists. Although different genes have been identified for their possible role in various asthma drug response profiles, their number remains small [42]. A selection of genes has been shown to be associated with ICS response according to lung function, airway hyperresponsiveness, or presence of exacerbations. A polymorphism of the corticotropin-releasing hormone receptor 1 (CRHR1) has been related to improved lung function during ICS treatment [43], instead, polymorphisms in the TBX21 gene are likely to be associated to an improvement in airway hyperresponsiveness [44]. Other genes are involved in ICS response such as GLCC11 according to the lung function [44] or FCER2 according to the exacerbations rate [45].

In this scenario, subjective predilections and aims of treatment remain a relevant tool in selecting the most appropriate treatment for each specific patient. Moreover, the constant verification and re-assessment of the correct use of inhaler devices is also particularly relevant. Similar to what happens in adulthood, the control of compliance and verification of adherence is a central aspect in the management of the disease, and in relation to the involvement of the caregiver figure, often performed by parents.

The Global Initiative for Asthma (GINA) Strategy Report reserves to clinicians a specific updated evidence-based strategy for asthma management in children [46,47]. In children ≥6 years, it is strongly suggested to confirm the diagnosis of asthma before starting controller treatment; instead, in children ≤5 years, a clinical diagnosis of asthma can be made when they suffer of wheezing or coughing with exercise/laughing/crying in the absence of respiratory infections, or if they have a history of eczema or allergic rhinitis. ICS-containing treatment is recommended in children 6–11 years with asthma instead in children ≤5 years, inhaled SABA is initially recommended in the management of wheezing and controller therapy can be performed if the symptom pattern suggests asthma, differential diagnoses have been excluded, and respiratory symptoms and/or wheezing episodes are frequent or severe [15,16,48]. It is extremely important to share treatment choices with the child’s parent/guardian, making sure to explain the relative benefits and risks of treatment for the child’s physical and social development. The GINA Strategy underlines that the choice of inhaler device should be based on the child’s age and capability. The pressurized metered dose inhaler with spacer is generally preferred, with face mask for children younger than 3 years, and a mouthpiece for most aged 3–5 years, and also in this population, before stepping up treatment, the clinician must exclude any modifiable risk factors including incorrect inhaler technique. At any age, moreover, the importance of providing regular training in inhaler technique and asthma self-management, even when symptoms are infrequent, is stressed. Furthermore, in relation to the COVID-19 pandemic, the use of nebulization could be reserved for life-threatening asthma, considering that bronchodilators administered by pressurized metered dose inhalers (pMDI) via valved holding chambers (VHC) are clinically at least as effective as nebulizers [49,50]. Different types of dispensers for lung deposition of drugs are available: nebulizers, pressurized inhalers (pMDIs) and dry powder inhalers (DPls), and soft mist inhalers (SMI). Several factors can guide the choice for a specific patient, especially in children. The Italian Society of Pediatrics, in line with the GINA recommendations, suggests for asthma management that specific inhaler devices should be used in different age groups: children, aged 4–6, should be treated by pMDI + spacer, while in children aged 6–12, there is no significant difference between pMDI and DPls [51,52]. It must be underlined that many asthmatic children use their inhalers devices too poorly to result in reliable drug delivery, even after inhalation instruction, and according to these data, comprehensive inhalation instructions and repeated check-ups are needed to achieve a reliable inhalation technique [53,54].

Validated and reproducible biomarkers could help in learning asthma pathways and discovering new possible therapeutic targets in asthma. Biomarkers should be potentially helpful in treatment selection and monitoring, particularly towards the use of new and expensive biological drugs, in predicting exacerbations or function decline. Where a single complete identifying biomarker is missing, the use of multiple biomarkers such as FeNO, sputum eosinophilia, exhaled breath condensate, IgE, or metabolites represents an important attempt at phenotyping patients even in childhood or adolescence [1,55].

Single or even pooled biomarkers have had limited impact in predicting clinical course or therapeutic efficacy in children [56].

Moreover, as regards the response to therapies, pediatric patients with severe asthma constitute a specific subpopulation. Although severe asthma affects approximately 4% to 5% of children with asthma, it accounts for the major portion of the morbidity, mortality, and cost of asthma [57]. According to data of this large longitudinal study, some pivotal aspects of children with severe asthma appear likely to be relevant: a half of children with severe asthma no longer qualified as having severe asthma 3 years later, asthma severity decreased equally in male and female subjects during longitudinal follow-up in adolescence, children with a greater peripheral eosinophil showed an increased odds of no longer having severe asthma after 3 years, and neither low lung function nor high BMI were observed in the children whose asthma severity classification did not improve [57].

Taken together all these observations identify the major unmet needs in the understanding and management of both asthma and severe asthma in childhood and adolescence that need to be systematically investigated through future studies.

## 7. Adolescents, Asthma, and Sleep Disorders

Adolescence is a period of transition to adulthood [58] and involves changes that affect personality, emotions, studies, social, and family life [59]. Although it is not yet clear the relative biological significance of sleep, in this phase, it seems crucial to the correct development of the formative period. Many studies indicate that adolescents sleep much less than required [60], and quantity and quality are further reduced in asthmatic patients.

The third International Classification of Sleep Disorders (ICSD-3) includes: insomnia; sleep-related breathing disorders, central disorders of hypersomnolence, circadian rhythm, sleep-wake disorders, sleep-related movement disorders, parasomnias, and other sleep disorders [61].

Some of these are very widespread, and this is why there is a high probability of coexistence with bronchial asthma, equally widespread, considered by the WHO (World Health Organization) the most frequent chronic disease in pediatric/adolescent age [62].

Sleep disorders and asthma affect each other (Figure 1). This applies especially to insomnia, respiratory sleep disorders, and circadian rhythm disorders. Insomnia disorder is characterized by dissatisfaction with sleep duration or quality and difficulties in initiating or maintaining sleep, which leads to a reduction in the sleeping time and its quality. Among all sleep disorders it represents the most frequent in the asthmatic adolescent [62,63] and, despite these findings, is underestimated, poorly characterized, little faced. Insomnia can be secondary to asthma and its therapies [64], be present independently [65], or both with the creation of a vicious circle that is self-sustaining with serious consequences on health and more (Figure 1).

Poorly controlled asthma, weight gain, depression, mental capacity reduction, tendency to illicit drug use, poor scholastic performance, and problems in social and familiar spheres, are just a few of the many consequences [66]. As mentioned before, adolescents are chronically sleep-deprived for physiological reasons (strong propensity to nocturnal chronotype–less homeostatic pressure–delay in melatonin production), which negatively impacts on social and educational commitments, the simultaneous presence of asthma aggravates the level of lack of sleep, and the use of electronic devices, especially in the evening, aggravates sleep deprivation [66].

Insomnia disorder in asthma, especially in uncontrolled asthma and more in severe asthma, identify a bi-directional mechanism: the presence of nocturnal symptoms interrupts the continuity of sleep, and the consequent insomnia favors the lack of control of the symptoms. In fact, when asthma is not controlled, nocturnal awakenings are frequent and linked to the need to take a drug [67]. Furthermore, the physiological vagal hypertonia, which occurs in sleep, represents a favorable condition for poor symptom control and the circadian rhythm affects the inflammatory state in asthma [67]. Although asthma therapies improve sleep, they may themselves cause insomnia; see here steroids, especially if taken under systemic administration, without ignoring the possible effects of inhaled corticosteroids (ICS) and their absorption, and in particular when taken in high doses. A stimulating effect is demonstrated for bronchodilators and antileukotriene, which, in addition to promoting sleepwalking and somniloquy (parasomnia NREM), also promote insomnia disorder [60,61]. Improving sleep quality and quantity is highly recommended to achieve asthma control in adolescents [62,63,64,65]. Cognitive Behavioral Therapy (CBT-I) for insomnia is highly recommended due to the positive outcomes observed in sleepless adolescents [68].

Studies indicate that OSA (Obstructive Sleep Apnea) is frequent even in pediatric asthmatic patients with variable outcomes depending on the diagnostic method. However, a high prevalence of the disorder emerges despite the improvement of asthma after an adenotonsillectomy [69]. Anatomical propensity in pediatrics is represented by hypertrophy of the tonsils and adenoids [70] associated with obesity favored by chronic use of ICS in moderate-severe asthma [71], by use of steroids [72], and by insomnia and sleep deprivation [73]. Obesity involves a chronic inflammatory condition from oxidative stress [47] enhanced by OSA with negative consequences in controlling asthma despite treatment [48,50,52,54]. It is necessary to research OSA even in asthmatic pediatric patients, so it is recommended to have a healthy lifestyle.

## 8. Conclusions

Children and adolescents represent an at-risk population to develop asthma due to poorly controlled conditions because of the coexistence of many factors. Climate change due to carbon emissions is impacting human lives and health across the globe, and from this point of view, may contribute to complicating the reaching of control in respiratory diseases such as asthma. Additionally, emotional factors such as anxiety and life stressors, diet, and lifestyle may contribute to worsening asthma control in this population. Even if their role needs to be better defined, a multidisciplinary approach that includes social workers/mental health professionals or dietician/nutritionist should be provided in difficult-to-treat cases. Finally, uncontrolled asthma and sleep disorders affect each other, contributing to globally decrease the quality of life in children and adolescents. 

## Figures and Tables

**Figure 1 children-09-00332-f001:**
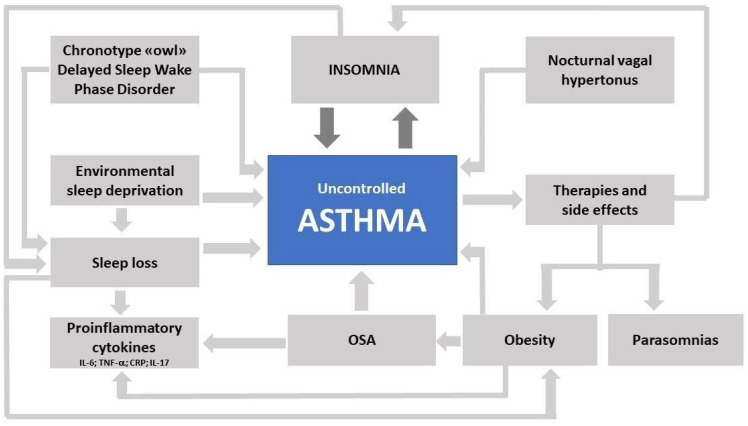
The effects of uncontrolled asthma on sleep disorders.

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
