# Peer review of "Heterogeneous Condition of Asthmatic Children Patients: A Narrative Review"

_children, 2022, doi:10.3390/children9030332_

Round 1

Reviewer 1 Report

This review provides a narrative review on heterogeneous condition of young asthmatic patients. It is well written and well organized. There are few things can be added:

1) In the transitional care section, would authors discuss how puberty (i.e. changes in sex hormones, growth spurt, or emotional stress) may affect the care of asthma in adolescents and young adults (AYA)

2) Please discuss how diet or medication (i.e. fiber intake, use of antibiotics) may play the role in the association between microbiome and airway inflammation 

3) In the conclusion, please address how clinicians should assess exposure to stressors (eg, violence or abuse) and screen for anxiety and depressive disorders, as well as emphasize in lifestyle change (healthier diet) when caring for young patients with asthma. In addition to providing referrals to social workers/mental health professionals or dietician/nutritionist when appropriate

Author Response

REVIEWER 1

This review provides a narrative review on the heterogeneous condition of young asthmatic patients. It is well written and well organized. There are few things can be added:

1) In the transitional care section, would authors discuss how puberty (i.e. changes in sex hormones, growth spurt, or emotional stress) may affect the care of asthma in adolescents and young adults (AYA)

A1. Thanks for raising this point. We have implemented the section underlining better the psychological challenges related to puberty (and all biological and behavioral changes)  that may deeply influence the adherence to treatment and therefore potentially induce a failure to take responsibility for self-management.

2) Please discuss how diet or medication (i.e. fiber intake, use of antibiotics) may play the role in the association between microbiome and airway inflammation 

R: Many thanks for your suggestion. However, we specifically focused on the association of the microbiota with the development of asthma. In the discussion, we highlighted the role of antibiotics as a detrimental factor and contributory cause of the disease. As for diet, evidence is still too poor to clearly identify a role in the pathogenesis of asthma in children.

3) In the conclusion, please address how clinicians should assess exposure to stressors (eg, violence or abuse) and screen for anxiety and depressive disorders, as well as emphasize in lifestyle change (healthier diet) when caring for young patients with asthma. In addition to providing referrals to social workers/mental health professionals or dietician/nutritionist when appropriate

R: Thank you for the suggestion, the text has been updated.

Reviewer 2 Report

Review of the article: Heterogeneous condition of young asthmatic patients: a narrative review

Title: I believe that the name “young asthmatic patients” could be replaced by children because the idea of “young asthmatics” suggests young children

In the introduction the authors select “overview” on adolescent asthma but in the article all children are included

The role of spirometry in asthma is lacking, is there any reason for it?

In page 3, 2nd paragraph a reference is lacking (XX instead of a reference number)

In page 4 , third paragraph “behavioral changes” related do covid/asthma crisis  reduction I  would add a comment of the use of masks as a potential cause for this (less viral transmition).

In the same page in the end “some trial  data even suggest” A reference should be included

In page 6 I suggest to add the importance of education including training and checking the inhaler use and sharing decisions with patients and their families)

In page 8  4th paragraph “even the use of electrical devises” should be replaced by electronic devices.

Author Response

REVIEWER 2

Title: I believe that the name “young asthmatic patients” could be replaced by children because the idea of “young asthmatics” suggests young children

 R: we appreciate the suggestion and have changed the title of the manuscript

In the introduction the authors select “overview” on adolescent asthma but in the article all children are included

R: thank you for the clarification. We changed the title of the paragraph with “pediatric asthma” in relation to the entire age range including up to 18 years.

The role of spirometry in asthma is lacking, is there any reason for it?

R: Thanks for this comment. We deliberately focused the paragraph on the “Therapeutic approach to asthma and severe asthma in childhood and adolescence” to stay within the scope of this narrative review but in consideration of this we have underlined this lack in the paragraph in order to make the reader aware. Thank you very much for this reflection, and we believe that the problem of the use of spirometry in the management of pediatric asthma deserves an in-depth discussion, perhaps in a dedicated review. 

In page 3, 2nd paragraph a reference is lacking (XX instead of a reference number)

R:Thanks for the report. We have corrected the mistake

In page 4 , third paragraph “behavioral changes” related do covid/asthma crisis reduction I would add a comment of the use of masks as a potential cause for this (less viral transmission).

R: thank you for your useful clarification. We absolutely agree and we have underlined this aspect by inserting a specific reference

In the same page in the end “some trial  data even suggest” A reference should be included

R: we have inserted it into the text

In page 6 I suggest to add the importance of education including training and checking the inhaler use and sharing decisions with patients and their families)

R: Thank you for pointing out this lack. We have proceeded to insert a specific comment on this aspect

In page 8  4th paragraph “even the use of electrical devices” should be replaced by electronic devices.

R: thank you for your suggestion. we have changed the term